# The Effect of the Supplementation of a Diet Low in Calcium and Phosphorus with Either Sheep Milk or Cow Milk on the Physical and Mechanical Characteristics of Bone using A Rat Model

**DOI:** 10.3390/foods9081070

**Published:** 2020-08-07

**Authors:** Keegan Burrow, Wayne Young, Niels Hammer, Sarah Safavi, Mario Scholze, Michelle McConnell, Alan Carne, David Barr, Malcolm Reid, Alaa El-Din Bekhit

**Affiliations:** 1Department of Wine, Food and Molecular Biosciences, Lincoln University, PO Box 85084, Lincoln 7647, New Zealand; 2AgResearch Ltd, Grasslands Research Centre, Private Bag 11008, Manawatu Mail Centre, Palmerston North 4442, New Zealand; Wayne.Young@agresearch.co.nz; 3Riddet Institute, Massey University, Massey University, Private Bag 11222, Palmerston North 4442, New Zealand; 4Department of Macroscopic and Clinical Anatomy, Medical University of Graz, 8010 Graz, Austria; niels.hammer@medunigraz.at; 5Department of Orthopaedic and Trauma Surgery, University of Leipzig, Liebigstraße 20, 04103 Leipzig, Germany; 6Fraunhofer IWU, Nöthnitzer Straße 44, 01187 Dresden, Germany; 7Department of Anatomy, University of Otago, PO Box 56, Dunedin 9054, New Zealand; sarah_safavi@gmx.de; 8Institute of Materials Science and Engineering, Chemnitz University of Technology, Straße der Nationen, 62, 09111 Chemnitz, Germany; mario.scholze@mb.tu-chemnitz.de; 9Department of Microbiology and Immunology, University of Otago, PO Box 56, Dunedin 9054, New Zealand; michelle.mcconnell@otago.ac.nz; 10Department of Biochemistry, University of Otago, PO Box 56, Dunedin 9054, New Zealand; alan.carne@otago.ac.nz; 11Department of Chemistry, University of Otago, PO Box 56, Dunedin 9054, New Zealand; dbarr@chemistry.otago.ac.nz (D.B.); malcolm.reid@otago.ac.nz (M.R.); 12Department of Food Science, University of Otago, PO Box 56, Dunedin 9054, New Zealand

**Keywords:** bone, deficient diet, dairy, minerals, micro-computed tomography, inductively coupled plasma–mass spectrometry, mechanical properties, sheep, cow

## Abstract

This study assessed the effect of cow milk (CM) and sheep milk (SM) consumption on the micro-structure, mechanical function, and mineral composition of rat femora in a male weanling rat model. Male weanling rats were fed a basal diet with a 50% reduction in calcium and phosphorus content (low Ca/P-diet) supplemented with either SM or CM. Rats were fed for 28 days, after which the femora were harvested and stored. The femora were analyzed by μ-CT, three-point bending, and inductively coupled plasma–mass spectrometry (ICP-MS). The addition of either milk to the low Ca/P-diet significantly increased (*p* < 0.05) trabecular bone volume, trabecular bone surface density, trabecular number, cortical bone volume, and maximum force, when compared to rats that consumed only the low Ca/P-diet. The consumption of either milk resulted in a significant decrease (*p* < 0.05) in trabecular pattern factor, and cortical bone surface to volume ratio when compared to rats that consumed only the low Ca/P-diet. The results were achieved with a lower consumption of SM compared to that of CM (*p* < 0.05). This work indicates that SM and CM can help overcome the effects on bone of a restriction in calcium and phosphorus intake.

## 1. Introduction

Deviation from adequate nutrition can result in changes in growth rates and exert wider biological effects including bone development in both humans and animals [1,2]. Ca deficiency is common worldwide [3]. As established by Gaddi et al. [4], even in relatively affluent countries such as Italy, about 36% of the population have shown signs of Ca imbalance. It is well established that dietary Ca and P deficiency has a direct impact on bone development. Work by Govindarajan et al. [5] investigated the effects of dietary Ca and P deficiency, by 85% and 93% respectively, on the mechanical and structural composition of ovariectomized female rat femora. The authors reported a significant reduction (*p* < 0.05) in trabecular bone volume (BV/TV (%)), measured by µ-CT) and break load (N, measured by three-point bend testing) after 3 months, compared to animals fed a control diet. Lobo et al. [1] studied the effects of a 72% reduction in Ca intake compared with the control diet, in 4-week-old male Wistar rats for a period of 33 days. In this work a significant reduction (*p* < 0.05) in BV/TV, break load (N), and Ca (mmol/g dry weight) were identified.

Although the use of dietary supplements to remedy nutritional imbalances is popular, the use of food rather than supplements to ensure adequate nutrition should be prioritized [4,6]. The use of Ca supplements have been associated with increased risk of age-related macular degeneration, and myocardial infarction [7,8]. Traditionally, the role of cow milk (CM) as an important source of Ca and P has been the focus of literature [9]. Milk from different species is known to have differences in both macro- and micro-composition and therefore to have the potential to act differently than CM with respect to nutrition [9,10]. Previous work from our lab has shown significant improvements in trabecular bone surface to volume ratio (BS/BV) and bone surface density (BS/TV) due to the consumption of sheep milk (SM) compared to the consumption of CM in a male weanling rat model (*p* < 0.05) [11]. The rats in the animal model used in Burrow et al. [11] were provided with a balanced diet that ensured an intake of both Ca and phosphorus (P) that was more than the recommended daily intake. Therefore, it is important to consider the effects of SM and CM as part of the diet rather than in addition to it.

In addition to the effect of mineral intake on bone mechanical and physical properties, the absorption and deposition of essential and trace minerals in bone may play an important role in supporting bone health [10]. In particular, the role and effects of non-radioactive (stable) isotope trace minerals on bone health has not been well established in the literature [12,13]. Thus, the full impact of milk on bone properties requires the monitoring of a broad spectrum of minerals.

The study aimed to assess the effect of SM and CM supplementation on the micro-structure, mechanical function, and mineral composition of the femora from male weanling rats fed a Ca and P restricted diet. In the present study, we hypothesize that the supplementation of milk to a Ca and P restricted diet will improve the nutrition of a deficient diet and that sheep milk will be more effective than CM in achieving this goal.

## 2. Materials and Methods

The design of the experiment was reviewed by a biometrician, taking into account the maximization of statistical power. In addition, an unbalanced experimental design was prepared in order to reduce the number of animals required, taking into account previously published works [11,14]. As required under the New Zealand Animal Welfare Act 1999 [15], all experiments conducted in this work involving animals were independently reviewed and approved by the AgResearch Grasslands Animal Ethics Committee (Palmerston North, New Zealand) (application number 14440).

### 2.1. Animals

All experimental procedures relating to animal handling and feeding have been previously reported in Burrow et al. [16]. Male Sprague–Dawley rats, three- to four-weeks of age (newly weaned) were sourced from the Hercus Taieri Resource Unit (University of Otago, Dunedin, New Zealand). During the trial, the rats were housed at the AgResearch small animal facility (Palmerston North, New Zealand).

### 2.2. Diets and Procedures

The diet groups for this experiment were the same as those described in Burrow et al. [16] as follows: 1) control (modified-AIN-93M) diet (control, *n* = 9); 2) low calcium and phosphorus with water (low Ca/P + water, *n* = 9); 3) low calcium and phosphorus with sheep milk (low Ca/P + SM, *n* = 15); and 4) low calcium and phosphorus with cow milk (low Ca/P + CM, *n* = 15. From arrival at the facility until the end of the feeding period, the rats were weighed approximately every 24 h.

Every two to three days fresh diet pellets (either a modified-AIN-93M or a low Ca/P modified-AIN-93M as indicated above) were provided to the rats (Table 1). The mass of pellets consumed by each rat was recorded and rejected pellets were discarded. Two times each day (once in the morning and once in the afternoon) the rats in the low Ca/P + SM and low Ca/P + CM groups were provided with whole un-pasteurized milk. Fresh drinking water was provided to the control and the low Ca/P + water animals *ad libitum*.

All rats were euthanized by standard procedures after 28 days of feeding using CO_2_ asphyxiation followed immediately by cervical dislocation as described in Burrow et al. [16]. The right femur of each rat was collected. The right femora were stripped of flesh manually and then wiped with gauze to remove as much remaining adherent tissue as possible. The femora were subsequently wrapped in gauze that had been soaked in physiological saline solution and frozen at −20 °C. 

### 2.3. Diet and Milk Composition

The macro-composition of the basal diets and milk samples has been reported previously in Burrow et al. [16]. Supplier information was used to determine the macro-composition of the basal diets. For the milk samples the macro-composition was determined using a Milkoscan™ (Foss Milkoscan, Foss, Hillerød, Denmark) applying the MilkTestNZ™ (Hamilton, New Zealand) standardized CM program.

### 2.4. Rat Femur Dimensions

The dimensions of the femur samples collected were measured using a 0 to 150 mm digital caliper (Mitutoyo, Sakado, Saitama, Japan). The total length and minimum shaft width were recorded.

### 2.5. µ-CT Analysis of Femora

The method used for assessing the given femora was adapted from McKinnon [17] and Bouxsein et al. [18] as reported by Burrow et al. [11]. The rat femora were scanned in a custom fixture using a Skyscan 1172 system (Bruker-Micro CT, Kontich, Belgium). Operating parameters used in the scanning process are provided in Appendix A. Bones were scanned from their distal end to the mid-point of the shaft.

NRecon software (Version 1.6.10.2) (Bruker-MicroCT, Kontich, Belgium) was used to reconstruct the image slices. The CTAn software (Version 1.14.4.1) (Bruker-MicroCT, Kontich, Belgium) was used for all remaining image processing and analysis. Images were binarized and the regions for analysis were isolated by interpolation of operator-drawn regions. The regions and the parameters assessed are as reported by Burrow et al. [11].

### 2.6. Three-Point Bend Testing of Femur Mechanical Properties

The femora were mechanically tested using a method adapted from Leppänen et al. [19] and McKinnon [17] using a Z020 device (Zwick & Roell, Ulm, Germany) equipped with a 2.5 kN load cell (Zwick & Roell). The bones were placed in the jig in a constant orientation by marking the mid-point of the shaft. The protocol included a fixed span width of 12 mm and a constant displacement rate of 10 mm/min.

### 2.7. Inductively Coupled Plasma–Mass Spectrometry (ICP-MS) Analysis of Femur Mineral Composition

Samples were digested in an ultraclean and metal-free Class 10 (ISO4) laboratory (Department of Chemistry, University of Otago, Dunedin, New Zealand). The method used for femur preparation, digestion and ICP-MS analysis were the same as Burrow et al. [11] (adapted from Raffalt et al. [20]).

#### 2.7.1. Femur Preparation

To remove any remaining adherent soft tissue, the femora were soaked sequentially in hydrogen peroxide (30% (*v*/*v*), analytical reagent grade), ethanol (95% (*v*/*v*), analytical reagent grade), and deionized water (18.2 MΩ.cm at 23.1 °C) for 1 h, 30 min and 30 min, respectively. The femora were then crushed using a pestle and mortar before being freeze-dried for a minimum of 24 h.

#### 2.7.2. Femur Digestion

Microwave digestion was conducted using a MARS 6 microwave digestion unit (CEM Corporation, Matthews, NC, USA) with Mars X-Press Teflon digestion tubes (CEM Corporation, Matthews, NC, USA). The microwave settings used were a ramp of 15 min and a hold of 15 min, with a temperature of 200 C, a pressure of 800 psi, and a power setting of 900–1050 W. Triple quartz distilled fuming nitric acid, 30% (v/v) hydrogen peroxide (analytical reagent grade; Labserv, Auckland, New Zealand) and deionized water (18.2 MΩ.cm at 23.1 °C) were used during this process.

The analysis was carried out using an ICP-MS (Agilent 7900, Agilent, Santa Clara, CA, USA) instrument in general purpose plasma mode and with a quartz 2.4 mm torch. The sample depth was set to 10 mm, with a gas flow rate of 1.05 L/min and a nebulizer flow rate of 0.1 rps. Internal standards of beryllium (Be), scandium (Sc), germanium (Ge), rhodium (Rh), indium (In), terbium (Tb) and bismuth (Bi) were added online. Detection limits are reported in Appendix A.

#### 2.7.3. Statistical Analyses

Statistical analysis was carried out using SPSS 24 (IBM Corporation, New York, NY, USA). For the comparison of diets where normal distribution was established, analysis of variance (ANOVA) was carried out followed by Tukey’s post hoc testing. For non-normally distributed data, the Kruskal–Wallis test and Dunn’s test with Bonferroni correction was used. Normal distribution was determined visually using P-P and Q-Q plots (generated in SPSS 24 (IBM Corporation, New York, NY, USA)). When normality was not confirmed, the data were handled as non-normally distributed.

## 3. Results

The diet intakes, composition, mineral intake per day, and growth rates of the rats have been previously reported in Burrow et al. [16].

### 3.1. Rat Femur Dimensions 

The dimensional measurements of the rat femora are shown in Figure 1. Higher bone mass and total lengths were found in the femora from the rats fed both milk-supplemented diets compared to the femora from the rats fed the low Ca/P + water diet (*p* < 0.05, Figure 1). In all cases, the femora from rats fed control diet were found to be not significantly different from the femora from rats fed any of the other three diets (*p* > 0.05, Figure 1).

### 3.2. µ-CT Analysis of Femur Micro-Structure

Significantly lower trabecular BV/TV, trabecular BS/TV, trabecular number (Tb.N), and cortical bone volume (BV) were observed in the femora from rats fed the low Ca/P + water diet compared to the femora from other diet groups (*p* < 0.05, Table 2 and Table 3). Significantly higher values were identified for trabecular pattern factor (Tb.Pf) and cortical BS/BV for the femora from rats fed the low Ca/P + water diet compared to the femora from rats in any of the other diet groups (*p* < 0.05, Table 2). For trabecular thickness (Tb.Ts), it was identified that the rat femora from both milk diets (low Ca/P +SM and low Ca/P +SM) had significantly lower values than the femora from rats fed the low Ca/P + water (*p* < 0.05, Table 2).

### 3.3. Three-Point Bending Testing of Femur Mechanical Properties

The results of the three-point bend testing of femora are reported in Table 4. It was found that femora from rats fed the low Ca/P diet had a significantly lower maximum force (F_max_) when compared with the femora from any other diet groups (*p* < 0.05, Table 4). With respect to strain at maximum force (ε at F_max_), it was found that femora from rats fed the control diet had significantly lower values than the two milk-containing diets (*p* < 0.05, Table 4). No significant difference was observed in the femora between the low Ca/P + water diet and the other diet groups for ε at F_max_ (*p* > 0.05, Table 4).

### 3.4. ICP-MS Analysis of the Femur Mineral Composition

Differences in the mineral composition of rat femora were identified due to the difference in the rat diet (Table 5). Significantly higher concentrations of Ca were present in the femora from rats fed the low Ca/P + SM and the control diet when compared with the femora from rats fed the low Ca/P + water diet (*p* < 0.05, Table 5). With respect to Fe, it was found that significantly higher concentrations were present in the femora from rats fed the low Ca/P + water diet when compared to the femora from rats fed the low Ca/P + SM diet (*p* < 0.05, Table 5). The highest concentration of P observed in the femora was for the low Ca/P + SM diet fed rats, and this was found to be significantly higher than the P concentration in the femora from rats fed the low Ca/P + water diet (*p* < 0.05, Table 5). No significant differences in the femur concentrations of P were observed when both the control and the low Ca/P + CM diets were compared to the femora from rats fed either of the other two diets (*p* > 0.05, Table 5). The low Ca/P + SM diet resulted in a significant increase in the concentration of Zn in the rat femora when compared with the femora from both the control and the low Ca/P + water fed rats (*p* < 0.05, Table 5).

It was found that both milk diets significantly increased the concentration of rubidium (Rb) in rat femora compared to the femora from rats fed the control diets (*p* < 0.05, Table 5). Sr was found to be significantly lower in the femora from the control fed rats than the femora from the rats fed any of the other three diets (*p* < 0.05, Table 5). Furthermore, the rats fed the low Ca/P + CM diet were found to have significantly lower concentrations of Sr in their femora, compared to that of the femora from rats fed either the low Ca/P + SM or the low Ca/P + water diets (*p* < 0.05, Table 5).

## 4. Discussion

### 4.1. Ca/P Deficient Diet Resulted in Significant Reductions in Structural and Mechanical Characteristics

The significantly lower bone structural and mechanical characteristics in the femora from rats fed low Ca/P + water compared to the femora of rats fed control (Figure 1, Table 2 and Table 3) shows that a reduction in Ca and P intake by 50% was sufficient to achieve a measurable difference in aspects of bone development. This is important as although within the literature a reduction in Ca intake has been shown to reduce bone mechanical and physical properties, these effects were typically observed at higher levels of Ca restriction. For example Lobo et al. [1] reported that a reduction in dietary Ca by 72%, compared with the control diet, showed a significant reduction in bone mass (*p* < 0.05), but did not result in a significant reduction in bone length (*p* > 0.05) of male Wistar rats after 33 days of feeding. Likewise, Viguet-Carrin et al. [21] reported that rats fed a diet containing 0.2% Ca had a significantly lower (*p* < 0.05) bone mineral density (BMD), cortical area, total area, BV/TV, Tb.N, and Tb.Th values compared to rats fed a diet containing 0.5% Ca. The reduction in Ca that was used in the work of Viguet-Carrin et al. [21] represented a decrease in the Ca content of the diet of around 60%; however, no data on specific Ca intake were provided by the authors. There are a handful of studies that show a similar level of Ca restriction compared to our study, for example, Agata et al. [22]. Agata et al. [22] showed that a Ca intake of below 0.3% (representing a 50% decrease in comparison to the control) resulted in a significant reduction in BMD of the tibia. Our study is unique in the context of the literature due to a restriction of P intake, which did not occur in any of the above studies [1,21,22]. When the role of P is considered, we are unable to establish from our data any direct role. When compared to the work of Lobo et al. [1], Viguet-Carrin et al. [21] and Agata et al. [22], our data do not indicate that there has been any additional reduction in bone structural and mechanical characteristics. This is likely due to the way in which the Ca to P ratio was kept constant within the diet used for this study. It has been shown that alterations of the Ca:P ratio, and specifically reducing P intake while not reducing Ca intake can have a large effect on bone development [22,23,24]. It has been proposed that this is due to a Ca–P ratio-dependent transport system [25].

### 4.2. Impact of Sheep Milk and Cow Milk on Bone Structural and Mechanical Characteristics

The results presented with respect to femur dimensions (Figure 1), femur micro-structure (Table 2 and Table 3), and femur mechanical characteristics show that the consumption of either SM or CM by the rats resulted in a recovery of macro-structural, micro-structural, and mechanical characteristics when compared to the low Ca/P + water fed rats. Comparing the femur properties of rats fed the control, low Ca/P + SM, and low Ca/P + CM diets, there is no significant effect on femur bone mass (*p* > 0.05, Figure 1). This is even though the low Ca/P + SM, and low Ca/P + CM fed rats showed a significantly higher intake of both Ca and P than the control fed rats as previously reported in Burrow et al. [16]. These results indicate that “peak bone mass” in the femora, in relation to the age of the rats, was attained by the rats in all of these diet groups (control, low Ca/P + SM, and low Ca/P + CM). “Peak bone mass” is the theoretical maximum bone growth that can be achieved by a given individual. “Peak bone mass” can be typically indicated by bone macro-structural characteristics and mechanical properties. Once “peak bone mass” is obtained, any improvement in diet will have no practical effect on the further development of bone [26,27].

Separately from the concept of “peak bone mass”, there is a strong link between Ca intake and bone micro-structure, independent of the Ca source in the diet [3]. The consistent effects observed between the low Ca/P + SM, and low Ca/P + CM fed rats, with respect to bone micro-structure are as expected. Previous studies have identified minor effects of different milk, dairy products, or dairy product derivatives on bone micro-structure while simultaneously identifying no changes in bone macro-structure or mechanics [3]. For example Fried et al. [28] showed that the consumption of skim milk powder (at 2.4 % of the total diet as part of a high-fat/high-sucrose diet) was able to significantly increase (*p* < 0.05) trabecular BV/TV, Tb.N, Tb.Ts, and trabecular BS/BV in bones in obese male Sprague–Dawley rats when compared with rats that were fed casein (at 2.4 % of the total diet as part of a high-fat/high-sucrose diet). Similar effects have also been observed when comparing SM and CM in the work of Burrow et al. [11], where significant improvements in BS/BV and BS/TV were observed due to the consumption of SM compared to the consumption of CM in a male weanling rat model (*p* < 0.05), but no differences occurred in bone mechanical properties. In the work of Burrow et al. [11] the rats that consumed SM had a significantly higher intake of Ca and P when compared to the rats that consumed CM. However, in the present work, the overall Ca and P intakes for the rats from the milk-containing diet groups (low Ca/P + SM and low Ca/P + CM) were not statistically different from each other (*p* > 0.05) as previously reported in Burrow et al. [16]. However, the rats that consumed the low Ca/P + SM diet consumed significantly less milk than the rats fed the low Ca/P + CM diet (*p* < 0.05) (as reported in Burrow et al. [16]). This is an important consideration as it means that the same femur dimensions and femur micro-structure can be obtained following the consumption of either SM or CM, but significantly less SM was consumed by the rats in achieving this when compared to CM.

### 4.3. Impact of Diet on Femur Mineral Composition

The Ca and P mineral concentrations present in the femora (Table 5) were consistent with the dietary intakes for each respective mineral for each respective diet when compared with the dietary intake data as reported in Burrow et al. [16]. Furthermore, the patterns with respect to the concentrations observed, reflect that of the data obtained for femur dimensions, femur micro-structure, and femur mechanical properties.

Fe deficiency, specifically iron deficient anemia, has been associated with a decrease in bone development [29,30]. In our work, a significant decrease in Fe concentration was observed in femora from rats fed the low Ca/P + SM diet when compared with femora from rats fed the low Ca/P + water diet (Table 5). No specific measurement was taken to determine anemic status of the rats in this trial. It must be noted that when inducing ferropenic anemia in rats, a substantial reduction in Fe intake, and/or absorption is required. For example, Campos et al. [31] provided rats with a diet containing 5 mg/kg of Fe. In our study the basal diets contained 44.7 and 41.8 mg/kg of Fe (for the modified-AIN-93M and low Ca/P modified-AIN-93M diets, respectively) (as reported in Burrow et al. [16]). The reduction in the Fe concentration in the femora of the low Ca/P + SM fed rats did not translate into any reduction in the micro-structure or mechanical properties of the femora when compared to any other diet group (Table 2, Table 3 and Table 4). This indicates that although a reduction in Fe accumulation was observed in femora from rats fed the low Ca/P + SM diet, this reduction was not sufficient to alter the bone properties.

Zn has been found to play a critical role in bone development, and this has been characterized as both a direct relationship with the differentiation process of osteoblasts, as well as the subsequent function of these cells [32,33]. In the present work, an inverse association between Zn accumulation in the femora (Table 5) and Zn intake (as reported in Burrow et al. [16]) can be observed. Specifically, this is that both milk-containing diets resulted in higher concentrations of Zn in the femora than either the control or the low Ca/P + water diets, with the low Ca/P + SM fed rats having a significantly higher Zn concentration in the femora (*p* < 0.05, Table 5). In previous studies, such as that of Ovesen et al. [34], a reduction in Zn in the diet (from 47 mg/kg to 2.042 mg/kg) significantly reduced bone mechanical characteristics including femur maximum load (N), in weanling male Wistar rats fed over a 3 week period (*p* < 0.05). Simultaneously (in the same work). It must be noted that both specific Zn intake and Zn absorption (and or accumulation) were not reported by Ovesen et al. [34], which makes it difficult to compare with the data presented in this work. The increased Zn accumulation in the femora from rats fed the low Ca/P + SM diet did not result in changes in the femur micro-structure and mechanical properties (Table 2, Table 3 and Table 4). This is because significant differences were observed in the Zn accumulation in the femora between the low Ca/P + SM and control fed rats, while simultaneously no significant differences were observed between the same two diets with respect to these characteristics (micro-structure and mechanical properties). Our data when compared to that of Ovesen et al. [34] indicate a form of matrix effect whereby the SM composition is able to increase the deposition of Zn in the femora. However, further work would be required to confirm this and elucidate any specific mechanisms.

Sr is unique within the data presented in Table 5. This is because it is the only mineral to show a significantly lower concentration in the femora for the control group in comparison to the two milk diets. As a divalent cation, it is believed that Sr has a limited capacity to replace Ca within the structure of bone [35,36]. This might justify why the femora of the low Ca/P + water fed rats showed statistically similar levels of Sr as that of the rats fed the low Ca/P + SM diet, as the femora of the low Ca/P + water fed rats Sr could act to mitigate the effects of Ca restriction. In turn, the increased level of Sr in both the milk fed groups when compared to the control group is most likely due to the increased level of Sr in the diet (as reported in Burrow et al. [16]). Sr is not considered to be either an essential or a trace mineral. Some evidence indicates that Sr has a role in bone health. It is believed that an increase in Sr may improve both mechanical and micro-structural properties in bone simultaneously, whereas a lack of Sr is not believed to have negative impacts [37]. In the present study, no effects on the femora resulting from the accumulation of Sr in the femora were identified with respect to structural or mechanical properties (Table 2, Table 3 and Table 4). What is particularly problematic is that most of the work associated with the effects of Sr on bone health is focused on the medical use of Sr, specifically strontium ranelate and strontium citrate [37,38]. More work is required with respect to the role of Sr derived from the diet.

When the mineral accumulation data in the present study are taken collectively, it can be observed that the consumption of either SM or CM can restore the negative effects of the consumption of a low Ca and P diet in rat femora. These negative effects are a reduction in bone mass (g), trabecular BV/TV, trabecular BS/TV, Tb.N, BV, and F_max_ (*p* < 0.05). The benefits of SM and CM are driven by the changes in Ca and P intake, and subsequent accumulation of these minerals rather than the effects of other minerals such as Fe, Zn, and Sr. In the case of Sr, specifically more research within the context of food products is needed. No unique effect of either milk type (CM or SM) has been identified with respect to the mineral accumulation in the femora.

## 5. Conclusions

The results obtained from this trial show that the consumption of either SM or CM, in addition to the consumption of a low Ca and P diet, results in the same bone mass, trabecular BV/TV, trabecular BS/TV, Tb.N, BV, and F_max_ in the femora of rats, as does the consumption of the control diet (*p* > 0.05). This was most likely due to the dietary Ca and P being provided by the milk.

Although the consumption of milk in addition to the Ca and P restricted diet was identified to significantly (*p* < 0.05) reduce the accumulation of Fe in the femora, this phenomenon was not found to alter the micro-structural parameters, mechanical properties, or mineral accumulation within the femora. This was also true for the increase in Rb concentration observed in the femora of rats that consumed milk. Significantly higher concentrations of Sr were identified in the femora from rats fed SM (*p* < 0.05). However, before any conclusions can be drawn from these data, further investigation is required with respect to the role of dietary Sr in bone development, specifically in the context of the diet rather than in any medical application.

It must be noted that as this work investigates the role of either SM or CM, in addition to the consumption of a low Ca and P diet in a rat model, further investigation is required to establish the effects over a longer period of time, and to determine if the effects observed in this study translate to humans.

## Figures and Tables

**Figure 1 foods-09-01070-f001:**
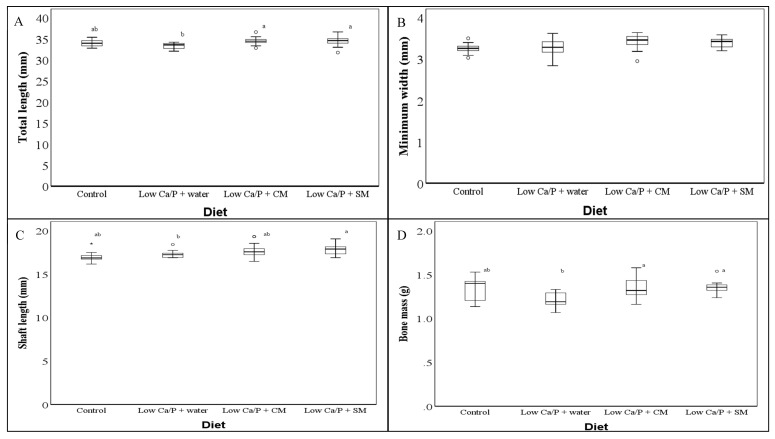
(**A–D**): Macroscopic dimensions of the femora of rats fed different diets ° and * Indicate outliers as identified by SPSS 24. Significant differences in bone parameters between the different diets are indicated by superscript letters as determined by the Kruskal–Wallis test and Dunn’s test with Bonferroni correction (*p* < 0.05).

**Table 1 foods-09-01070-t001:** Ingredients (g/kg) of the diet pellets used in the animal feeding trial (Reproduced from Burrow et al. [16]) *.

Diet	Modified-AIN-93M(g/kg)	Low Ca/P Modified-AIN-93M (g/kg)
Ingredient		
Corn starch	495.69	499.27
Beef protein extract	140	141
Maltodextrin	125	125.9
Sucrose	106.69	107.5
Cellulose	50	50
Soybean oil	40	40
CaCO_3_ ^a^	12.495	6.293
Vitamin mix ^b^	10	10
KH_2_PO_4_ ^c^	8.75	4.407
Mineral mix ^d^	3.5	3.5
NaCl	2.59	2.61
Choline bitartrate	2.5	2.5
L-cystine	1.8	1.8
K₃C₆H₅O ^e^	0.98	4.41
t-butylhydroquinone	0.008	0.008
Yellow dye ^f^	0	0.05

* Values as provided by the supplier, additional detail on the mineral composition of the diet is available in Appendix A. ^a^ 40% Ca. ^b^ Proprietary mix (V10037). ^c^ 22.8% P and 28.7% K. ^d^ Proprietary mix (S10022C). ^e^ 36.2% K. ^f^ Tartrazine.

**Table 2 foods-09-01070-t002:** Trabecular bone characteristics of rat femora from rats fed different diets.

Diet		Control	Low Ca/P + water	Low Ca/P + CM	Low Ca/P + SM
Parameter	Unit				
BV/TV	%	45.2 ± 3.81 ^a^	41.6 ± 2.78 ^b^	45.2 ± 3.03 ^a^	44.7 ± 3.63 ^a^
BS/BV	mm^2^/mm^3^	33.6 ± 2.33	34.0 ± 2.10	33.2 ± 1.65	33.8 ± 1.99
BS/TV	mm^2^/mm^3^	15.1 ± 1.53 ^a^	14.1 ± 0.88 ^b^	15.0 ± 0.90 ^a^	15.1 ± 0.97 ^a^
Tb.Th	mm	0.09 ± 0.004	0.09 ± 0.003	0.09 ± 0.003	0.09 ± 0.003
Tb.Ts	mm	0.27 ± 0.06 ^ab^	0.30 ± 0.04 ^a^	0.26 ± 0.03 ^b^	0.26 ± 0.03 ^b^
Tb.N.	1/mm	5.15 ± 0.47 ^a^	4.75 ± 0.27 ^b^	5.09 ± 0.32 ^a^	5.10 ± 0.36 ^a^
Tb.Pf	N/A	−19.9 ± 3.34 ^b^	−16.0 ± 2.86 ^a^	−18.9 ± 2.65^b^	−18.9 ± 3.25 ^b^

Results are presented as mean ± standard deviation, significant differences in bone parameters between the different diets are indicated by superscript letters as determined by ANOVA and Tukey’s post hoc testing (*p* < 0.05), Low Ca/P = low calcium and phosphate, SM = sheep milk, CM = cow milk, BV/TV = bone volume, BS/TV = bone surface density, BS/BV = bone surface to volume ratio, Tb.Pf = trabecular pattern factor, Tb.Th = Trabecular thickness, Tb.Ts = Trabecular separation, Tb.N =Trabecular number, refer to Appendix A for parameter descriptions, N/A = Dimensionless measurement.

**Table 3 foods-09-01070-t003:** Cortical bone characteristics of femora from rats fed different milk diets.

Diet		Control	Low Ca/P + water	Low Ca/P + CM	Low Ca/P + SM
Parameter	Unit				
BV	mm^3^	14.2 ± 1.83 ^a^	12.7 ± 1.24 ^b^	14.7 ± 1.32 ^a^	14.8 ± 1.56 ^a^
BS	mm^2^	95.3 ± 20.8	93.8 ± 13.4	96.6 ± 9.08	98.6 ± 12.3
BS/BV	mm^2^/mm^3^	6.70 ± 0.97 ^b^	7.42 ± 1.03 ^a^	6.59 ± 0.51 ^b^	6.69 ± 0.75 ^b^
PV	mm^3^	0.03 ± 0.03	0.02 ± 0.01	0.02 ± 0.01	0.03 ± 0.01
BSD	mm^2^/mm^3^	0.33 ± 0.10	0.32 ± 0.07	0.32 ± 0.07	0.32 ± 0.07
P%	%	0.19 ± 0.16	0.15 ± 0.12	0.13 ± 0.08	0.17 ± 0.09

Results are presented as mean ± standard deviation, significant differences in bone parameters between the different diets are indicated by superscript letters as determined by ANOVA and Tukey’s post hoc testing (*p* < 0.05), BV = Bone volume, BS = Bone surface, BS/BV = Bone surface to volume ratio, PV = Volume of pores, BSD = Bone surface density, P% = Percentage porosity, refer to Appendix A for parameters descriptions.

**Table 4 foods-09-01070-t004:** Mechanical characteristics of femora from rats fed different milk diets.

Diet		Control	Low Ca/P + water	Low Ca/P + CM	Low Ca/P + SM
Parameter	Unit				
E_mod_	GPa	2.26 ± 0.65	1.93 ± 0.65	1.82 ± 0.90	1.87 ± 0.42
F_max_	N	103 ± 17.6 ^a^	93.1 ± 12.9 ^b^	103 ± 13.1 ^a^	107 ± 17.0 ^a^
F_break_	N	65.2 ± 25.6	61.1 ± 33.1	57.6 ± 25.6	64.2 ± 27.5
σ_max_	N/mm²	97.7 ± 22.3	86.9 ± 23.2	84.4 ± 34.3	89.9 ± 14.7
dl at F_max_	mm	0.67 ± 0.14	0.71 ± 0.12	0.66 ± 0.10	0.73 ± 0.18
dl at break	mm	1.02 ± 0.20	1.17 ± 0.24	1.19 ± 0.20	1.11 ± 0.20
ε at F_max_	%	9.06 ± 1.95 ^b^	9.66 ± 1.73 ^ab^	9.42 ± 1.44 ^a^	10.3 ± 2.73 ^a^

Results are presented as mean ± standard deviation, superscript letters indicate significant differences in bone parameters between the different diets as determined by ANOVA and Tukey’s post hoc testing (*p* < 0.05), tested using a three-point bend test (2.5 kN load cell, a fixed span of 12 mm and a constant depression rate of 10 mm/min), E_Mod_ = Elastic modulus (Young’s modulus), F_max_ = Maximum force, F_break_ = Force at break, σ_max_ = Maximum tensile stress, dl at F_max_ = Deflection and maximum force, dl at break = Deflection at break, ε at F_max_ = Strain at maximum force.

**Table 5 foods-09-01070-t005:** Mineral concentrations in the femora of rats fed different diets.

Diet	Control (mg/kg)	Low Ca/P + water (mg/kg)	Low Ca/P + CM (mg/kg)	Low Ca/P + SM (mg/kg)
Element				
Al	45.8 ± 16.9	40.1 ± 31.6	33.6 ± 19.5	31.2 ± 17.8
Ca ^#^	227 ± 4.20 ^a^	218 ± 7.69 ^b^	221 ± 7.80 ^ab^	225 ± 8.14 ^a^
Cu	0.78 ± 0.07	0.86 ± 0.23	0.82 ± 0.08	0.79 ± 0.11
Fe	35.0 ± 9.82 ^ab^	40.5 ± 10.6 ^a^	31.4 ± 7.34 ^ab^	29.8 ± 10.2 ^b^
K	224 ± 104	256 ± 42.3	289 ± 47.6	273 ± 47.0
Mg ^#^	3.90 ± 0.33	3.76 ± 0.33	3.77 ± 0.21	3.80 ± 0.24
Mn	0.54 ± 0.25	0.68 ± 0.04	0.63 ± 0.05	0.63 ± 0.06
Na ^#^	5.03 ± 0.08	4.83 ± 0.22	5.08 ± 0.21	4.99 ± 0.33
P ^#^	116 ± 2.93 ^ab^	112 ± 3.82 ^b^	114 ± 4.09 ^ab^	116 ± 3.75 ^a^
Pb	0.31 ± 0.28	0.75 ± 0.37	0.67 ± 0.79	0.95 ± 0.67
Rb	0.15 ± 0.08 ^b^	0.19 ± 0.03 ^b^	0.29 ± 0.06 ^a^	0.27 ± 0.056 ^a^
Sr	53.3 ± 7.19 ^c^	82.5 ± 17.4 ^a^	67.2 ± 13.7 ^b^	85.0 ± 9.72 ^a^
Zn	152 ± 67.8 ^b^	176 ± 18.6 ^b^	184 ± 15.9 ^ab^	195 ± 12.4 ^a^

Results are presented as mean ± standard deviation, significant differences in bone parameters between the different diets are indicated by superscript letters as determined by the Kruskal–Wallis test and Dunn’s test with Bonferroni correction (*p* < 0.05). ^#^ [g/kg].

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
