# Peer review of "The Effect of the Supplementation of a Diet Low in Calcium and Phosphorus with Either Sheep Milk or Cow Milk on the Physical and Mechanical Characteristics of Bone using A Rat Model"

_foods, 2020, doi:10.3390/foods9081070_

Round 1
Reviewer 1 Report
This article examines the effects of supplementing calcium and phosphorus deficient diets with sheep's milk (SM) and cow's milk (CM) on bone mechanical properties and mineral content of femurs in rats. The authors report that SM and CM both provide significant benefits to bone despite Ca/P mineral deficiencies in the diet. Overall, the study is fairly novel and gives some interesting ideas to consider regarding SM and CM as supplement to the diet. However, given that dairy products are the main source of calcium in the diet (and phosphorus is typically consumed in excess in modern diets), the relevance of a diet deficient in both (and then supplemented with dairy) seems to be a convoluted way of testing the hypothesis that milk provides calcium for bone health.
There are a number of aspects to this article that are interesting and would be helpful to expand our knowledge on the topic. However, as currently constructed, the authors do not successfully convey the novelty of their study/results and fail to convincingly present their data in a way that gives readers a streamlined scientific story. The writing quality is poor and this further confuses the presentation of results. As currently constructed, this article is in need of considerable revision and rewriting.
Introduction - The justification for this project is incomplete. Is this a paper focused on nutrition and bone strength? If so, further justification and explanation of the explicit connections between diet (specifically calcium and phosphorus) and bone mechanical properties is needed. In addition to citing several studies (which the authors do), a more comprehensive view of the topic is necessary. Later sections of this manuscript focus extensively on other minerals, so why are those not mentioned here? This needs to be justified/explained in intro so reader knows why you are pursuing so many elements. Additionally, the last two paragraphs of intro are poorly constructed and do not give adequate explanation or justification for the study that is being introduced. Why SM and CM? Why not calcium dietary supplement? Why just Ca and P deficient diets? There is not nearly enough justification for why the current experiment is being done or why the topic is of interest.
Methods - Overall, this section is well constructed. It is clearly and concisely written with enough detail for reader to follow procedures and experiment. However, table 1 needs improvement. What are units? Is this g/kg diet? Additionally, the "mineral mix" may impact calcium and phos content of diet and should be explicitly stated. I see footnote that it is "proprietary mix", but simple ICP experiment can determine exact Ca and P content of diet. The totals of each diet are different (i.e., columns add up to different total amounts of diet), and the CaCO3, KH2PO4, K3C6H5O are likely the discrepency. This needs to be checked and modified. Table 2 should be moved to Supporting Info.
Results - This section needs substantial improvement in writing quality, figure quality, and justification for measuring minerals. The writing is very convoluted and uses far too many words to state what different groups are compared to other groups and then what different parameters are within each group (confusing, right?). The writing needs to be shortened and the main differences more clearly highlighted for the reader. As currently constructed, I'm not sure what the meaningful differences observed in this study actually are. Section 3.1 and Table 3/4 are directly copied from ref. 10; there's no need to reproduce them here. Additionally, many data are shown in both table and figure form. This is a redundant repetition of the data. Select the most important data to highlight visually and include that in 1-2 high quality figures. All other data can be presented in tables, but there is no reason to duplicate all data.
Discussion - This section is far too long and doesn't offer the reader much in the way of interpretation and importance of results. The results are repeated ad nauseam, which is entirely unnecessary. There are several other studies from the literature cited, but the results are not fully integrated and contextualized with the current study. Why are those other studies important and why are you citing them? How do your data line up (or not) with those studies? When taken together, how do your results and other studies in the literature lead you to a meaningful conclusion and interpretation of your results? And, what are the implications of that conclusion? Sentences like lines 354-356 are good, but then the remainder of that paragraphs goes backwards when it could point to conclusion. Section 4.3 discussed iron, strontium, and zinc, but neglect calcium and phosphorus. This doesn't make sense, as Ca and P were altered in diet and are (presumably) main factors in SM/CM that impact bone strength. While discussing Fe/Sr/Zn is ok, these are minor factors and take a back seat to Ca/P, which are dietary drivers of bone changes observed here.
Mineral intakes and femoral content - Many minerals were measured in this study, but this seems like a fishing expedition. That is, the authors measured every mineral they could on the ICP and then looked at results to see if anything was significant/meaningful. For example, why do we care how much erbium, yttrium, neodymium, or uranium is in the diets? These are not meaningful minerals and, quite frankly, are most likely indicative of contamination or background environmental levels. If all these minerals are going to be measured, then some justification must be given for the importance of these minerals and why they were measured.
Figures - Quality needs to be drastically improved. The figures are poorly constructed and very hard to read. Bigger font sizes, fewer lines, clearer stats, and less blurry figures are imperative to being able to clearly read and interpret data. Additionally, figure captions are very repetitive and long, but do not offer meaningful information regarding what is in figure. These need to be rewritten to make figures clear to reader. (The same applies to table footnotes, as the same info does not need to be repeated in each; it's just padding and extending the length unnecessarily.)
Author Response
We thank the Reviewer for their comments which have helped improve out manuscript. Please see the attachment for a full breakdown of the revisions made.

Reviewer 2 Report
I’ve read with attention the paper of Burrow et al. that is potentially of interest. The study assessed the effect of cow milk and sheep milk consumption on the micro-structure, mechanical function, and mineral composition of rat femora in a male weanling rat model. The paper is well-written and easy to read. The background and aim of the study have been clearly defined. The methodology applied is overall correct, I only suggested to stress the potential limitation of the proposed methodology and the future research perspectives on this subject. Moreover, I suggested the authors to reference the paper Eur J Clin Nutr. 2001 Feb;55(2):97-106. doi: 10.1038/sj.ejcn.1601123. Compared with other published material, the study suggest that the use of milk can balance the effect of a calcium- and posphours poor diet on bone metabolism (at least, in the rat). The conclusion are consistent with the new data produced by the authors. And address the main question posed.
Author Response

(The authors gave the same response as above.)

Round 2
Reviewer 1 Report
The revised version of this manuscript represents substantial improvement from the initial draft. Specifically, the methods, results, tables, and figure are all of good quality. However, several significant concerns remain regarding the introduction and discussion/conclusion.
The introduction, while offering more information and justification than the previous draft, still lacks focus in creating a coherent argument for the need to do this study. For example, in lines 53-57, the authors cite studies regarding excess calcium intake. Then, in the following lines, they cite deficient calcium intake. These are two different issues and, as the current study deals only with Ca/P deficient diet, focusing on that comparison alone is most relevant to the work presented here. That said, additional focus on the effects of Ca/P deficiency would help strengthen the argument for conducting this study.
Lines 71-72 presents a fact that doesn't flow with the argument being made. Perhaps the point is that affluent countries are more likely to consume CM? Clarification is necessary if this sentence is to be included.
Lines 73-92 present good information, but it is poorly written and I'm struggling to follow the logic and reasoning here. I appreciate the addition of information regarding minerals and bone, but the writing needs to clearly emphasize why these facts are presented and how they logically lead to the hypothesis presented.
The motivation for this study could be stated more clearly, and the introduction should lead the reader to easily understand what this motivation is. As currently constructed, this is NOT obvious to reader.
Line 82 - I recommend moving hypothesis to last paragraph of introduction.
Lines 85-92 - The bioavailability and matrix effects of different food sources are important factors to consider. However, given that there was nothing in the present study to address this issue, I'm not sure how it fits into the introduction? Streamlining the introduction to focus in on the key issues motivating this study is key to guiding the reader.
Section 2.2 - how much milk was provided to rats? how much SM and CM was consumed? Several times throughout the study, it is mentioned that it took less SM than CM to achieve same bone results. Is this because they drank less SM? Clarification would be appreciated... Also in this section, define "standard procedures" for euthanasia.
It is confusing to have two groups called "control". I would recommend renaming the "low Ca/P + control" group as simply "low Ca/P" or "low Ca/P + water" to avoid confusion with the other control group. This will also help clarify the results and discussion sections when comparisons between groups are made.
Figure 1C - Control group has * symbol in graph. What does this mean?
Tables - Define all annotations (^, *, #, etc.) in footnotes.
Discussion - This section is improved, but still needs a lot of work before being ready to publish. Sections 4.2 and 4.3 are far too long and don't give the reader much in the way of interpretation of results. There are a number of good sources cited, however, as currently constructed, much of this section reads like a narrative review article (where other studies are described and summarized) rather than a discussion section of a primary research article (where the results of the current study are interpreted in the context of the broader field). To be clear, the studies cited are relevant, but the details provided from each study are not necessary. Focus on what you did and why it's novel and interesting; use other literature to help give context to that. For example, lines 326-329 make an interesting statement. Expand on this and discuss this type of information more throughout the discussion. These are the important points for readers to understand, but you must lead the readers to these points and then expand upon them.
The conclusion makes several good points, though it can be shortened to about half its current length and avoid repetition.
Author Response
We thank the Reviewer for their comments which have helped improve out manuscript. Please see the Please see the attachment for the detailed responses
